# A* Lasso for Learning a Sparse Bayesian Network Structure for Continuous Variables

**Jing Xiang**
Machine Learning Department
Carnegie Mellon University
Pittsburgh, PA 15213
jingx@cs.cmu.edu

**Seyoung Kim**
Lane Center for Computational Biology
Carnegie Mellon University
Pittsburgh, PA 15213
sssykim@cs.cmu.edu

## Abstract

We address the problem of learning a sparse Bayesian network structure for continuous variables in a high-dimensional space. The constraint that the estimated Bayesian network structure must be a directed acyclic graph (DAG) makes the problem challenging because of the huge search space of network structures. Most previous methods were based on a two-stage approach that prunes the search space in the first stage and then searches for a network structure satisfying the DAG constraint in the second stage. Although this approach is effective in a low-dimensional setting, it is difficult to ensure that the correct network structure is not pruned in the first stage in a high-dimensional setting. In this paper, we propose a single-stage method, called A* lasso, that recovers the optimal sparse Bayesian network structure by solving a single optimization problem with A* search algorithm that uses lasso in its scoring system. Our approach substantially improves the computational efficiency of the well-known exact methods based on dynamic programming. We also present a heuristic scheme that further improves the efficiency of A* lasso without significantly compromising the quality of solutions. We demonstrate our approach on data simulated from benchmark Bayesian networks and real data.

## 1 Introduction

Bayesian networks have been popular tools for representing the probability distribution over a large number of variables. However, learning a Bayesian network structure from data has been known to be an NP-hard problem [1] because of the constraint that the network structure has to be a directed acyclic graph (DAG). Many of the exact methods that have been developed for recovering the optimal structure are computationally expensive and require exponential computation time [15, 7]. Approximate methods based on heuristic search are more computationally efficient, but they recover a suboptimal structure. In this paper, we address the problem of learning a Bayesian network structure for continuous variables in a high-dimensional space and propose an algorithm that recovers the exact solution with less computation time than the previous exact algorithms, and with the flexibility of further reducing computation time without a significant decrease in accuracy.

Many of the existing algorithms are based on scoring each candidate graph and finding a graph with the best score, where the score decomposes for each variable given its parents in a DAG. Although methods may differ in the scoring method that they use (e.g., MDL [9], BIC [14], and BDe [4]), most of these algorithms, whether exact methods or heuristic search techniques, have a two-stage learning process. In Stage 1, candidate parent sets for each node are identified while ignoring the DAG constraint. Then, Stage 2 employs various algorithms to search for the best-scoring network structure that satisfies the DAG constraint by limiting the search space to the candidate parent sets from Stage 1. For Stage 1, methods such as sparse candidate [2], max-min parents children [17], and

total conditioning [11] algorithms have been previously proposed. For Stage 2, exact methods based on dynamic programming [7, 15] and A* search algorithm [19] as well as inexact methods such as heuristic search technique [17] and linear programming formulation [6] have been developed. These approaches have been developed primarily for discrete variables, and regardless of whether exact or inexact methods are used in Stage 2, Stage 1 involved exponential computation time and space.

For continuous variables, $L_1$-regularized Markov blanket (L1MB) [13] was proposed as a two-stage method that uses lasso to select candidate parents for each variable in Stage 1 and performs heuristic search for DAG structure and variable ordering in Stage 2. Although a two-stage approach can reduce the search space by pruning candidate parent sets in Stage 1, Huang *et al.* [5] observed that applying lasso in Stage 1 as in L1MB is likely to miss the true parents in a high-dimensional setting, thereby limiting the quality of the solution in Stage 2. They proposed the sparse Bayesian network (SBN) algorithm that formulates the problem of Bayesian network structure learning as a single-stage optimization problem and transforms it into a lasso-type optimization to obtain an approximate solution. Then, they applied a heuristic search to refine the solution as a post-processing step.

In this paper, we propose a new algorithm, called A* lasso, for learning a sparse Bayesian network structure with continuous variables in high-dimensional space. Our method is a single-stage algorithm that finds the optimal network structure with a sparse set of parents while ensuring the DAG constraint is satisfied. We first show that a lasso-based scoring method can be incorporated within dynamic programming (DP). While previous approaches based on DP required identifying the exponential number of candidate parent sets and their scores for each variable in Stage 1 before applying DP in Stage 2 [7, 15], our approach effectively combines the score computation in Stage 1 within Stage 2 via lasso optimization. Then, we present A* lasso which significantly prunes the search space of DP by incorporating the A* search algorithm [12], while guaranteeing the optimality of the solution. Since in practice, A* search can still be expensive compared to heuristic methods, we explore heuristic schemes that further limit the search space of A* lasso. We demonstrate in our experiments that this heuristic approach can substantially improve the computation time without significantly compromising the quality of the solution, especially on large Bayesian networks.

## 2   Background on Bayesian Network Structure Learning

A Bayesian network is a probabilistic graphical model defined over a DAG $G$ with a set of $p = |V|$ nodes $V = \{v_1, \ldots, v_p\}$, where each node $v_j$ is associated with a random variable $X_j$ [8]. The probability model associated with $G$ in a Bayesian network factorizes as $p(X_1, \ldots, X_p) = \prod_{j=1}^{p} p(X_j | \mathrm{Pa}(X_j))$, where $p(X_j | \mathrm{Pa}(X_j))$ is the conditional probability distribution for $X_j$ given its parents $\mathrm{Pa}(X_j)$ with directed edges from each node in $\mathrm{Pa}(X_j)$ to $X_j$ in $G$. We assume continuous random variables and use a linear regression model for the conditional probability distribution of each node $X_j = \mathrm{Pa}(X_j)'\boldsymbol{\beta}_j + \epsilon$, where $\boldsymbol{\beta}_j = \{\beta_{jk}\text{'s for } X_k \in \mathrm{Pa}(X_j)\}$ is the vector of unknown parameters to be estimated from data and $\epsilon$ is the noise distributed as $N(0, 1)$.

Given a dataset $\mathbf{X} = [\mathbf{x}_1, \ldots, \mathbf{x}_p]$, where $\mathbf{x}_j$ is a vector of $n$ observations for random variable $X_j$, our goal is to estimate the graph structure $G$ and the parameters $\boldsymbol{\beta}_j$'s jointly. We formulate this problem as that of obtaining a sparse estimate of $\boldsymbol{\beta}_j$'s, under the constraint that the overall graph structure $G$ should not contain directed cycles. Then, the nonzero elements of $\boldsymbol{\beta}_j$'s indicate the presence of edges in $G$. We obtain an estimate of Bayesian network structure and parameters by minimizing the negative log likelihood of data with sparsity enforcing $L_1$ penalty as follows:

$$\min_{\boldsymbol{\beta}_1, \ldots, \boldsymbol{\beta}_p} \sum_{j=1}^{p} \| \mathbf{x}_j - \mathbf{x}_{-j}'\boldsymbol{\beta}_j \|_2^2 + \lambda \sum_{j=1}^{p} \| \boldsymbol{\beta}_j \|_1 \quad \text{s.t. } G \in \mathrm{DAG}, \tag{1}$$

where $\mathbf{x}_{-j}$ represents all columns of $\mathbf{X}$ excluding $\mathbf{x}_j$, assuming all other variables are candidate parents of node $v_j$. Given the estimate of $\boldsymbol{\beta}_j$'s, the set of parents for node $v_j$ can be found as the support of $\boldsymbol{\beta}_j$, $S(\boldsymbol{\beta}_j) = \{v_i | \beta_{ji} \neq 0\}$. The $\lambda$ is the regularization parameter that determines the amount of sparsity in $\boldsymbol{\beta}_j$'s and can be determined by cross-validation. We notice that if the acyclicity constraint is ignored, Equation (1) decomposes into individual lasso estimations for each node:

$$\mathrm{LassoScore}(v_j | V \backslash v_j) = \min_{\boldsymbol{\beta}_j} \| \mathbf{x}_j - \mathbf{x}_{-j}'\boldsymbol{\beta}_j \|_2^2 + \lambda \| \boldsymbol{\beta}_j \|_1,$$

where $V \backslash v_j$ represents the set of all nodes in $V$ excluding $v_j$. The above lasso optimization problem can be solved efficiently with the shooting algorithm [3]. However, the main challenge in optimizing Equation (1) arises from ensuring that the $\boldsymbol{\beta}_j$'s satisfy the DAG constraint.

## 3 A* Lasso for Bayesian Network Structure Learning

### 3.1 Dynamic Programming with Lasso

The problem of learning a Bayesian network structure that satisfies the constraint of no directed cycles can be cast as that of learning an optimal ordering of variables [8]. Once the optimal variable ordering is given, the constraint of no directed cycles can be trivially enforced by constraining the parents of each variable in the local conditional probability distribution to be a subset of the nodes that precede the given node in the ordering. We let $\Pi^V = [\pi_1^V, \dots, \pi_{|V|}^V]$ denote an ordering of the nodes in $V$, where $\pi_j^V$ indicates the node $v \in V$ in the $j$th position of the ordering, and $\Pi_{\prec v_j}^V$ denote the set of nodes in $V$ that precede node $v_j$ in ordering $\Pi^V$.

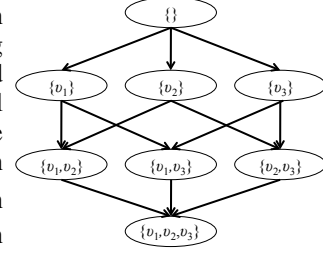

Figure 1: Search space of variable ordering for three variables $V = \{v_1, v_2, v_3\}$.

Algorithms based on DP have been developed to learn the optimal variable ordering for Bayesian networks [16]. These approaches are based on the observation that the score of the optimal ordering of the full set of nodes $V$ can be decomposed into (a) the optimal score for the first node in the ordering, given a choice of the first node and (b) the score of the optimal ordering of the nodes excluding the first node. The optimal variable ordering can be constructed by recursively applying this decomposition to select the first node in the ordering and to find the optimal ordering of the set of remaining nodes $U \subset V$. This recursion is given as follows, with an initial call of the recursion with $U = V$:

$$\text{OptScore}(U) = \min_{v_j \in U} \text{OptScore}(U \backslash v_j) + \text{BestScore}(v_j | V \backslash U) \qquad (2)$$

$$\pi_1^U = \operatorname*{argmin}_{v_j \in U} \text{OptScore}(U \backslash v_j) + \text{BestScore}(v_j | V \backslash U), \qquad (3)$$

where $\text{BestScore}(v_j | V \backslash U)$ is the optimal score of $v_j$ under the optimal choice of parents from $V \backslash U$.

In order to obtain $\text{BestScore}(v_j | V \backslash U)$ in Equations (2) and (3), for the case of discrete variables, many previous approaches enumerated all possible subsets of $V$ as candidate sets of parents for node $v_j$ to precompute $\text{BestScore}(v_j | V \backslash U)$ in Stage 1 before applying DP in Stage 2 [7, 15]. While this approach may perform well in a low-dimensional setting, in a high-dimensional setting, a two-stage method is likely to miss the true parent sets in Stage 1, which in turn affects the performance of Stage 2 [5]. In this paper, we consider the high-dimensional setting and present a single-stage method that applies lasso to obtain $\text{BestScore}(v_j | V \backslash U)$ within DP as follows:

$$\begin{aligned} \text{BestScore}(v_j | V \backslash U) &= \text{LassoScore}(v_j | V \backslash U) \\ &= \min_{\boldsymbol{\beta}_j, S(\boldsymbol{\beta}_j) \subseteq V \backslash U} \| \mathbf{x}_j - \mathbf{x}_{-j}' \boldsymbol{\beta}_j \|_2^2 + \lambda \| \boldsymbol{\beta}_j \|_1 . \end{aligned}$$

The constraint $S(\boldsymbol{\beta}_j) \subseteq V \backslash U$ in the above lasso optimization can be trivially maintained by setting the $\beta_{jk}$ for $v_k \in U$ to 0 and optimizing only for the other $\beta_{jk}$'s. When applying the recursion in Equations (2) and (3), DP takes advantage of the overlapping subproblems to prune the search space of orderings, since the problem of computing $\text{OptScore}(U)$ for $U \subseteq V$ can appear as a subproblem of scoring orderings of any larger subsets of $V$ that contain $U$.

The problem of finding the optimal variable ordering can be viewed as that of finding the shortest path from the start state to the goal state in a search space given as a subset lattice. The search space consists of a set of states, each of which is associated with one of the $2^{|V|}$ possible subsets of nodes in $V$. The start state is the empty set $\{\}$ and the goal state is the set of all variables $V$. A valid move in this search space is defined from a state for subset $Q_s$ to another state for subset $Q_{s'}$, only if $Q_{s'}$ contains one additional node to $Q_s$. Each move to the next state corresponds to adding a node at the end of the ordering of the nodes in the previous state. The cost of such a move is given by $\text{BestScore}(v | Q_s)$, where $v = Q_{s'} \backslash Q_s$. Each path from the start state to the goal state gives one

possible ordering of nodes. Figure 1 illustrates the search space, where each state is associated with a $Q_s$. DP finds the shortest path from the start state to the goal state that corresponds to the optimal variable ordering by considering all possible paths in this search space and visiting all $2^{|V|}$ states.

## 3.2 A* Lasso for Pruning Search Space

As discussed in the previous section, DP considers all $2^{|V|}$ states in the subset lattice to find the optimal variable ordering. Thus, it is not sufficiently efficient to be practical for problems with more than 20 nodes. On the other hand, a greedy algorithm is computationally efficient because it explores a single variable ordering by greedily selecting the most promising next state based on BestScore($v|Q_s$), but it returns a suboptimal solution. In this paper, we propose A* lasso that incorporates the A* search algorithm [12] to construct the optimal variable ordering in the search space of the subset lattice. We show that this strategy can significantly prune the search space compared to DP, while maintaining the optimality of the solution.

When selecting the next move in the process of constructing a path in the search space, instead of greedily selecting the move, A* search also accounts for the estimate of the future cost given by a heuristic function $h(Q_s)$ that will be incurred to reach the goal state from the candidate next state. Although the exact future cost is not known until A* search constructs the full path by reaching the goal state, a reasonable estimate of the future cost can be obtained by ignoring the directed acyclicity constraint. It is well-known that A* search is guaranteed to find the shortest path if the heuristic function $h(Q_s)$ is *admissible* [12], meaning that $h(Q_s)$ is always an underestimate of the true cost of reaching the goal state. Below, we describe an admissible heuristic for A* lasso.

While exploring the search space, A* search algorithm assigns a score $f(Q_s)$ to each state and its corresponding subset $Q_s$ of variables for which the ordering has been determined. A* search algorithm computes this score $f(Q_s)$ as the sum of the cost $g(Q_s)$ that has been incurred so far to reach the current state from the start state and an estimate of the cost $h(Q_s)$ that will be incurred to reach the goal state from the current state:

$$f(Q_s) = g(Q_s) + h(Q_s). \tag{4}$$

More specifically, given the ordering $\Pi^{Q_s}$ of variables in $Q_s$ that has been constructed along the path from the start state to the state for $Q_s$, the cost that has been incurred so far is defined as

$$g(Q_s) = \sum_{v_j \in Q_s} \text{LassoScore}(v_j | \Pi^{Q_s}_{\prec v_j}) \tag{5}$$

and the heuristic function for the estimate of the future cost to reach the goal state is defined as:

$$h(Q_s) = \sum_{v_j \in V \backslash Q_s} \text{LassoScore}(v_j | V \backslash v_j) \tag{6}$$

Note that the heuristic function is admissible, or an underestimate of the true cost, since the constraint of no directed cycles is ignored and each variable in $V \backslash Q_s$ is free to choose any variables in $V$ as its parents, which lowers the lasso objective value.

When the search space is a graph where multiple paths can reach the same state, we can further improve efficiency if the heuristic function has the property of *consistency* in addition to admissibility. A consistent heuristic always satisfies $h(Q_s) \leq h(Q_{s'}) + \text{LassoScore}(v_k | Q_s)$, where LassoScore($v_k | Q_s$) is the cost of moving from state $Q_s$ to state $Q_{s'}$ with $\{v_k\} = Q_{s'} \backslash Q_s$. Consistency ensures that the first path found by A* search to reach the given state is always the shortest path to that state [12]. This allows us to prune the search when we reach the same state via a different path later in the search. The following proposition states that our heuristic function is consistent.

**Proposition 1** *The heuristic in Equation (6) is consistent.*

**Proof** For any successor state $Q_{s'}$ of $Q_s$, let $v_k = Q_{s'} \backslash Q_s$.

$$h(Q_s) = \sum_{v_j \in V \backslash Q_s} \text{LassoScore}(v_j | V \backslash v_j)$$

$$= \sum_{v_j \in V \backslash Q_s, v_j \neq v_k} \text{LassoScore}(v_j | V \backslash v_j) + \text{LassoScore}(v_k | V \backslash v_k)$$

$$\leq h(Q_{s'}) + \text{LassoScore}(v_k | Q_s),$$

**Input** : $\mathbf{X}, V, \lambda$
**Output**: Optimal variable ordering $\Pi^V$
Initialize *OPEN* to an empty queue;
Initialize *CLOSED* to an empty set;
Compute LassoScore$(v_j|V\backslash v_j)$ for all $v_j \in V$;
*OPEN*.insert$((Q_s = \{\}, f(Q_s) = h(\{\}), g(Q_s) = 0, \Pi^{Q_s} = []))$;
**while** *true* **do**
   $(Q_s, f(Q_s), g(Q_s), \Pi^{Q_s}) \leftarrow$ *OPEN*.pop();
   **if** $h(Q_s) = 0$ **then**
      | Return $\Pi^V \leftarrow \Pi^{Q_s}$;
   **end**
   **foreach** $v \in V\backslash Q_s$ **do**
      $Q_{s'} \leftarrow Q_s \cup \{v\}$;
      **if** $Q_{s'} \notin$ *CLOSED* **then**
         Compute LassoScore$(v|Q_s)$ with lasso shooting algorithm;
         $g(Q_{s'}) \leftarrow g(Q_s) +$ LassoScore$(v|Q_s)$;
         $h(Q_{s'}) \leftarrow h(Q_s) -$ LassoScore$(v|V\backslash v)$;
         $f(Q_{s'}) \leftarrow g(Q_{s'}) + h(Q_{s'})$;
         $\Pi^{Q_{s'}} \leftarrow [\Pi^{Q_s}, v]$;
         *OPEN*.insert$(L = (Q_{s'}, f(Q_{s'}), g(Q_{s'}), \Pi^{Q_{s'}}))$;
         *CLOSED* $\leftarrow$ *CLOSED* $\cup \{Q_{s'}\}$;
      **end**
   **end**
**end**

**Algorithm 1:** A* lasso for learning Bayesian network structure

where LassoScore$(v_k|Q_s)$ is the true cost of moving from state $Q_s$ to $Q_{s'}$. The inequality above holds because $v_k$ has fewer parents to choose from in LassoScore$(v_k|Q_s)$ than in LassoScore$(v_k|V\backslash v_k)$. Thus, our heuristic in Equation (6) is consistent. ∎

Given a consistent heuristic, many paths that go through the same state can be pruned by maintaining an *OPEN* list and a *CLOSED* list during A* search. In practice, the *OPEN* list can be implemented with a priority queue and the *CLOSED* list can be implemented with a hash table. The *OPEN* list is a priority queue that maintains all the intermediate results $(Q_s, f(Q_s), g(Q_s), \Pi^{Q_s})$'s for a partial construction of the variable ordering up to $Q_s$ at the frontier of the search, sorted according to the score $f(Q_s)$. During search, A* lasso pops from the *OPEN* list the partial construction of ordering with the lowest score $f(Q_s)$, visits the successor states by adding another node to the ordering $\Pi^{Q_s}$, and queues the results onto the *OPEN* list. Any state that has been popped by A* lasso is placed in the *CLOSED* list. The states that have been placed in the *CLOSED* list are not considered again, even if A* search reaches these states through different paths later in the search.

The full algorithm for A* lasso is given in Algorithm 1. As in DP with lasso, A* lasso is a single-stage algorithm that solves lasso within A* search. Every time A* lasso moves from state $Q_s$ to the next state $Q_{s'}$ in the search space, LassoScore$(v_j|\Pi^{Q_s}_{\prec v_j})$ for $\{v_j\} = Q_{s'}\backslash Q_s$ is computed with the shooting algorithm and added to $g(Q_s)$ to obtain $g(Q_{s'})$. The heuristic score $h(Q_{s'})$ can be precomputed as LassoScore$(v_j|V\backslash v_j)$ for all $v_j \in V$ for a simple look-up during A* search.

### 3.3 Heuristic Schemes for A* Lasso to Improve Scalability

Although A* lasso substantially prunes the search space compared to DP, it is not sufficiently efficient for large graphs, because it still considers a large number of states in the exponentially large search space. One simple strategy for further pruning the search space would be to limit the size of the priority queue in the *OPEN* list, forcing A* lasso to discard less promising intermediate results first. In this case, limiting the queue size to one is equivalent to a greedy algorithm with a scoring function in Equation (4). In our experiments, we found that such a naive strategy substantially reduced the quality of solutions because the best-scoring intermediate results tend to be the results at the early stage of the exploration. They are at the shallow part of the search space near the start state because the admissible heuristic underestimates the true cost.

Instead, given a limited queue size, we propose to distribute the intermediate results to be discarded across different depths/layers of the search space. For example, given the depth of the search space

Table 1: Comparison of computation time of different methods

| Dataset (Nodes) | DP | A* lasso | A* Qlimit 1000 | A* Qlimit 200 | A* Qlimit 100 | A* Qlimit 5 | L1MB | SBN |
|---|---|---|---|---|---|---|---|---|
| Dsep (6) | 0.20 (64) | 0.14 (15) | – (–) | – (–) | – (–) | 0.17 (11) | 2.65 | 8.76 |
| Asia (8) | 1.07 (256) | 0.26 (34) | – (–) | – (–) | – (–) | 0.22 (12) | 2.79 | 8.9 |
| Bowling (9) | 2.42 (512) | 0.48 (94) | – (–) | – (–) | – (–) | 0.23 (13) | 2.85 | 8.75 |
| Inversetree (11) | 8.44 (2048) | 1.68 (410) | – (–) | 1.8 (423) | 1.16 (248) | 0.2 (16) | 3.03 | 8.56 |
| Rain (14) | 1216 (1.60e4) | 76.64 (2938) | 64.38 (1811) | 13.97 (461) | 7.88 (270) | 1.67 (17) | 12.26 | 10.19 |
| Cloud (16) | 1.6e4 (6.6e4) | 137.36 (2660) | 108.39 (1945) | 26.16 (526) | 9.92 (244) | 2.14 (19) | 4.72 | 14.56 |
| Funnel (18) | 4.2e4 (2.6e5) | 1527.0 (2.3e4) | 88.87 (2310) | 25.19 (513) | 11.53 (248) | 2.73 (21) | 4.76 | 10.08 |
| Galaxy (20) | 1.3e5 (1.0e6) | 2.40e4 (8.2e4) | 110.05 (3093) | 27.59 (642) | 12.02 (323) | 3.03 (23) | 6.59 | 11.0 |
| Factor (27) | – (–) | – (–) | 1389.7 (3912) | 125.91 (801) | 59.92 (397) | 3.96 (30) | 9.04 | 13.91 |
| Insurance (27) | – (–) | – (–) | 2874.2 (3448) | 442.65 (720) | 202.9 (395) | 16.31 (33) | 10.96 | 29.45 |
| Water (32) | – (–) | – (–) | 2397.0 (3442) | 301.67 (687) | 130.71 (343) | 12.14 (38) | 32.73 | 14.96 |
| Mildew (35) | – (–) | – (–) | 3928.8 (3737) | 802.76 (715) | 339.04 (368) | 29.3 (36) | 15.25 | 116.33 |
| Alarm (37) | – (–) | – (–) | 2732.3 (3426) | 384.87 (738) | 158.0 (378) | 12.42 (42) | 7.91 | 39.78 |
| Barley (48) | – (–) | – (–) | 10766.0 (4072) | 1869.4 (807) | 913.46 (430) | 109.14 (52) | 23.25 | 483.33 |
| Hailfinder (56) | – (–) | – (–) | 9752.0 (3939) | 2580.5 (816) | 1058.3 (390) | 112.61 (57) | 44.36 | 826.41 |

Table 2: A* lasso computation time under different edge strengths $\boldsymbol{\beta}_j$'s

| Dataset (Nodes) | (1.2,1.5) | (1,1.2) | (0.8,1) |
|---|---|---|---|
| Dsep (6) | 0.14 (15) | 0.14 (16) | 0.17 (30) |
| Asia (8) | 0.26 (34) | 0.23 (37) | 0.29 (59) |
| Bowling (9) | 0.48 (94) | 0.49 (103) | 0.54 (128) |
| Inversetree (11) | 1.68 (410) | 2.09 (561) | 2.25 (620) |
| Rain (14) | 76.64 (2938) | 66.93 (2959) | 97.26 (4069) |
| Cloud (16 ) | 137.36 (2660) | 229.12 (7805) | 227.43 (8858) |
| Funnel (18) | 1526.7 (22930) | 2060.2 (33271) | 3744.4 (40644) |
| Galaxy (20) | 24040 (82132) | 66710 (168492) | 256490 (220821) |

$|V|$, if we need to discard $k$ intermediate results, we discard $k/|V|$ intermediate results at each depth. In our experiments, we found that this heuristic scheme substantially improves the computation time of A* lasso with a small reduction in the quality of the solution. We also considered other strategies such as inflating heuristics [10] and pruning edges in preprocessing with lasso, but such strategies substantially reduced the quality of solutions.

## 4 Experiments

### 4.1 Simulation Study

We perform simulation studies in order to evaluate the accuracy of the estimated structures and measure the computation time of our method. We created several small networks under 20 nodes and obtained the structure of several benchmark networks between 27 and 56 nodes from the Bayesian Network Repository (the left-most column in Table 1). In addition, we used the tiling technique [18] to generate two networks of approximately 300 nodes so that we could evaluate our method on larger graphs. Given the Bayesian network structures, we set the parameters $\boldsymbol{\beta}_j$ for each conditional probability distribution of node $v_j$ such that $\beta_{jk} \sim \pm Uniform[l, u]$ for predetermined values for $u$ and $l$ if node $v_k$ is a parent of node $v_j$ and $\beta_{jk} = 0$ otherwise. We then generated data from each Bayesian network by forward sampling with noise $\epsilon \sim N(0, 1)$ in the regression model, given the true variable ordering. All data were mean-centered.

We compare our method to several other methods including DP with lasso for an exact method, L1MB for heuristic search, and SBN for an optimization-based approximate method. We downloaded the software implementations of L1MB and SBN from the authors' website. For L1MB, we increased the authors' recommended number of evaluations 2500 to 10 000 in Stage 2 heuristic search for all networks except the two larger networks of around 300 nodes (Alarm 2 and Hailfinder 2), where we used two different settings of 50 000 and 100 000 evaluations. We also evaluated A* lasso with the heuristic scheme with the queue sizes of 5, 100, 200, and 1000.

DP, A* lasso, and A* lasso with a limited queue size require a selection of the regularization parameter $\lambda$ with cross-validation. In order to determine the optimal value for $\lambda$, for different values of $\lambda$, we trained a model on a training set, performed an ordinary least squares re-estimation of the non-zero elements of $\boldsymbol{\beta}_j$ to remove the bias introduced by the $L_1$ penalty, and computed prediction errors on the validation set. Then, we selected the value of $\lambda$ that gives the smallest prediction error as the optimal $\lambda$. We used a training set of 200 samples for relatively small networks with under

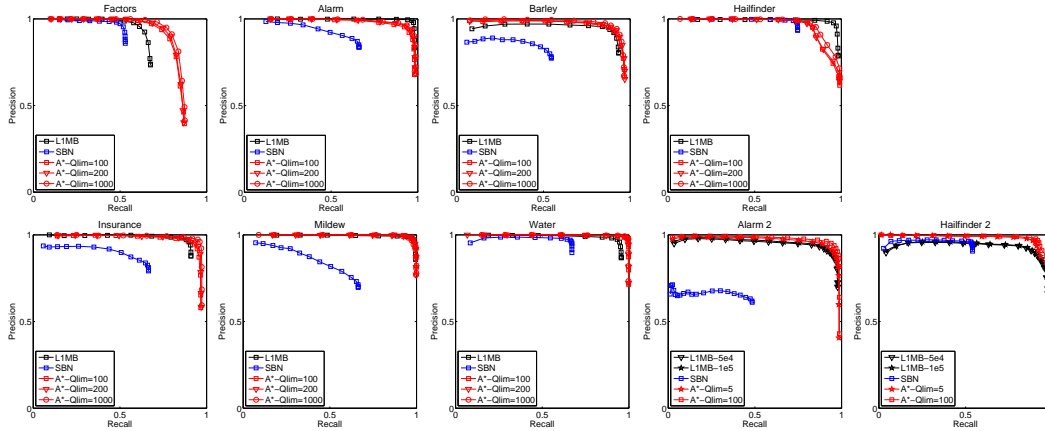

Figure 2: Precision/recall curves for the recovery of skeletons of benchmark Bayesian networks.

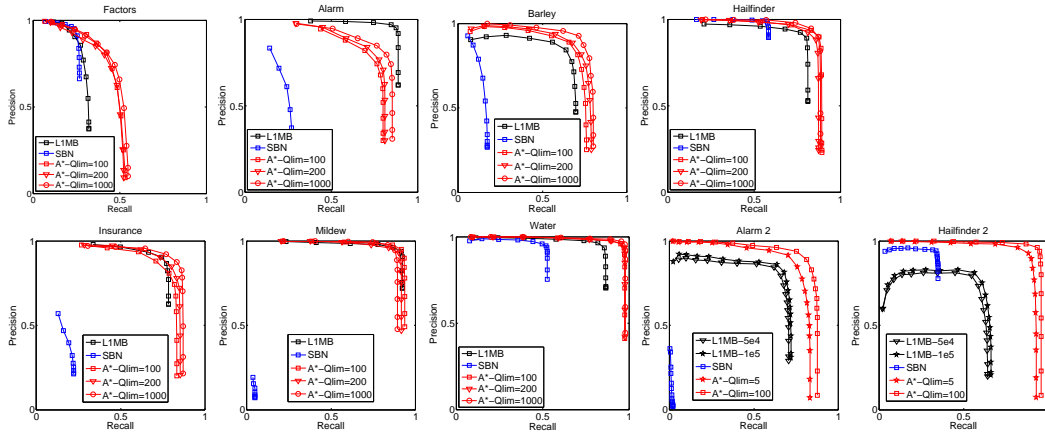

Figure 3: Precision/recall curves for the recovery of $v$-structures of benchmark Bayesian networks.

60 nodes and a training set of 500 samples for the two large networks with around 300 nodes. We used a validation set of 500 samples. For L1MB and SBN, we used a similar strategy to select the regularization parameters, while mainly following the strategy suggested by the authors and in their software implementation.

We present the computation time for the different methods in Table 1. For DP, A* lasso, and A* lasso with limited queue sizes, we also record the number of states visited in the search space in parentheses in Table 1. All methods were implemented in Matlab and were run on computers with 2.4 GHz processors. We used a dataset generated from a true model with $\beta_{jk} \sim \pm Uniform[1.2, 1.5]$. It can be seen from Table 1 that DP considers all possible states $2^{|V|}$ in the search space that grows exponentially with the number of nodes. It is clear that A* lasso visits significantly fewer states than DP, visiting about 10% of the number of states in DP for the funnel and galaxy networks. We were unable to obtain the computation time for A* lasso and DP for some of the larger graphs in Table 1 as they required significantly more time. Limiting the size of the queue in A* lasso reduces both the computation time and the number of states visited. For smaller graphs, we do not report the computation time for A* lasso with limited queue size, since it is identical to the full A* lasso. We notice that the computation time for A* lasso with a small queue of 5 or 100 is comparable to that of L1MB and SBN.

In general, we found that the extent of pruning of the search space by A* lasso compared to DP depends on the strengths of edges ($\boldsymbol{\beta}_j$ values) in the true model. We applied DP and A* lasso to datasets of 200 samples generated from each of the networks under each of the three settings for the true edge strengths, $\pm Uniform[1.2, 1.5]$, $\pm Uniform[1, 1.2]$, and $\pm Uniform[0.8, 1]$. As can be seen from the computation time and the number of states visited by DP and A* lasso in Table 2, as the strengths of edges increase, the number of states visited by A* lasso and the computation time tend to decrease. The results in Table 2 indicate that the efficiency of A* lasso is affected by the signal-to-noise ratio.

In order to evaluate the accuracy of the Bayesian network structures recovered by each method, we make use of the fact that two Bayesian network structures are indistinguishable if they belong to the same equivalence class, where an equivalence class is defined as the set of networks with the same skeleton and $v$-structures. The skeleton of a Bayesian network is defined as the edge connectivities ignoring edge directions and a $v$-structure is defined as the local graph structure over three variables, with two variables pointing to the other variables (i.e., $A \rightarrow B \leftarrow C$). We evaluated the performance of the different methods by comparing the estimated network structure with the true network structure in terms of skeleton and $v$-structures and computing the precision and recall.

The precision/recall curves for the skeleton and $v$-structures of the models estimated by the different methods are shown in Figures 2 and 3, respectively. Each curve was obtained as an average over the results from 30 different datasets for the two large graphs (Alarm 2 and Hailfinder 2) and from 50 different datasets for all the other Bayesian networks. All data were simulated under the setting $\beta_{jk} \sim \pm Uniform[0.4, 0.7]$. For the benchmark Bayesian

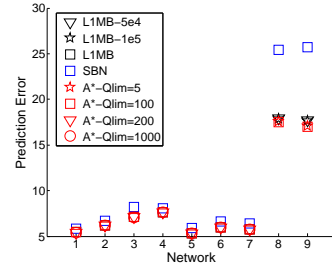

Figure 4: Prediction errors for benchmark Bayesian networks. The $x$-axis labels indicate different benchmark Bayesian networks for 1: Factors, 2: Alarm, 3: Barley, 4: Hailfinder, 5: Insurance, 6: Mildew, 7: Water, 8: Alarm 2, and 9: Hailfinder 2.

networks, we used A* lasso with different queue sizes, including 100, 200, and 1000, whereas for the two large networks (Alarm 2 and Hailfinder 2) that require more computation time, we used A* lasso with queue size of 5 and 100. As can be seen in Figures 2 and 3, all methods perform relatively well on identifying the true skeletons, but find it significantly more challenging to recover the true $v$-structures. We find that although increasing the size of queues in A* lasso generally improves the performance, even with smaller queue sizes, A* lasso outperforms L1MB and SBN in most of the networks. While A* lasso with a limited queue size preforms consistently well on smaller networks, it significantly outperforms the other methods on the larger graphs such as Alarm 2 and Hailfinder 2, even with a queue size of 5 and even when the number of evaluations for L1MB has been increased to 50 000 and 100 000. This demonstrates that while limiting the queue size in A* lasso will not guarantee the optimality of the solution, it still reduces the computation time of A* lasso dramatically without substantially compromising the quality of the solution. In addition, we compare the performance of the different methods in terms of prediction errors on independent test datasets in Figure 4. We find that the prediction errors of A* lasso are consistently lower even with a limited queue size.

## 4.2 Analysis of S&P Stock Data

We applied the methods on the daily stock price data of the S&P 500 companies to learn a Bayesian network that models the dependencies in prices among different stocks. We obtained the stock prices of 125 companies over 1500 time points between Jan 3, 2007 and Dec 17, 2012. We estimated a Bayesian network using the first 1000 time points with the different methods, and then computed prediction errors on the last 500 time points. For L1MB, we used two settings for the number of evaluations, 50 000 and 100 000. We applied A* lasso with different queue limits of 5, 100, and 200. The prediction accuracies for the various methods are shown in Figure 5. Our method obtains lower prediction errors than the other methods, even with the smaller queue sizes.

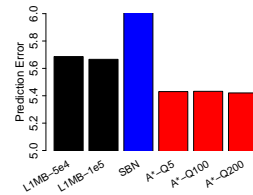

Figure 5: Prediction errors for S&P stock price data.

## 5 Conclusions

In this paper, we considered the problem of learning a Bayesian network structure and proposed A* lasso that guarantees the optimality of the solution while reducing the computational time of the well-known exact methods based on DP. We proposed a simple heuristic scheme that further improves the computation time but does not significantly reduce the quality of the solution.

### Acknowledgments

This material is based upon work supported by an NSF CAREER Award No. MCB-1149885, Sloan Research Fellowship, and Okawa Foundation Research Grant to SK and by a NSERC PGS-D to JX.

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
