[Reviews · NeurIPS 2013]

Submitted by Assigned_Reviewer_7

This paper introduces a method for finding Bayesian networks for continuous variables in high-dimensional spaces. The paper assumes a Gaussian distribution of any particular random variable when conditioned on its parent nodes. A LASSO objective function is used to construct a sparse set of parent nodes for each random variable, subject to an additional constraint that the resulting structure be an acyclic graph. The network structure constraint is framed as an ordering problem, and an A* search algorithm is proposed which finds a directed acyclic graph which maximizes the LASSO objective function. The LASSO objective function, minus the DAG constraint, is used as an admissible heuristic in the A* search. This search is still not fast enough for graphs with large numbers of nodes, so a method for further pruning the search space is introduced.

The paper is written clearly, with clear motivation and sufficient detail for implementation by a third party.

The use of the LassoScore for optimization in the dynamic programming algorithm, and using the unconstrained LassoScore as a heuristic function, is clever. The simulation studies adequately show that the approximate version of their algorithm still greatly outperforms conventional greedy approaches, while exhibiting fast runtimes even on large networks of nodes. The performance graphs indicate that even substantial pruning the search graph represents only minimal losses relative to the optimal solution.

While the paper describes the process as learning network structure for continuous variables, using a linear regression model with fixed noise error (section 2.1) for the conditional distributions would seem to suggest assuming a network in which each random variable is itself linear Gaussian. Offhand this appears to be a fairly strong assumption; however, as noted in the rebuttal, recovering structure is a difficult problem even in Gaussian networks.

The application described in section 3.1 is not very clear, and it is not described how stock price data is represented as a Bayesian network. This reviewer is not familiar with the use of Bayesian networks for stock market analysis, and it is not immediately clear to the uninitiated what sort of model is used, or why stock market data would exhibit a particular network structure. This is addressed by references in the rebuttal, which should be added to section 3.1.

Summary: This is a nice paper, which exploits the relationship between a DAG structure and an ordering of random variable dependencies to implement a clever A* algorithm for optimizing a constrained LASSO objective function; the resulting algorithm performs quickly and accurately, uncovering network structure in the synthetic data generated from this model.

Submitted by Assigned_Reviewer_8

The authors put forward a new class of approaches for learning sparse Bayesian Networks. Their first method is a DP approach that integrates the Lasso as a scoring method. Since this approach requires searching over a prohibitively large space, they modify it with an A* search (that still uses the Lasso scoring method). This method uses a heuristic function that is admissible and consistent (and thus is guaranteed to find the optimum and allows pruning). Their final method is an approximation of A* that works by limiting the queue size. While very small queues degrade the quality of solution, they find that for moderate limits, great speed improvements are possible with little degradation of quality.

The authors note that many other methods prune the search space in a first step and then find the best DAG in this reduced space as a separate step. This practice may exclude the optimum. Therefore, their approach seems like a good way to avoid doing this.

The experimental section seems pretty thorough.

This paper is well-written and well-organized.

In Figure 3, what was varied to generate the curves? Lambda? I ask because the non-monotonicity in the Hailfinder 2 plot is a little surprising. Also, does SBN really drop below the "random guessing" line in four places? This might indicate something fishy.

Minor comments:
- Line 21: Missing space: "two-stageapproach"
- Line 130: "U\in V" should be "U\subseteq V"
- Line 140: empty citation? []
- Lines 157 and 175: "and and"
- Line 169 and 171: Don't start both with "On the other hand"
- Line 180: heuristic misspelled
- Line 258: CLOSE should be CLOSED
- Line 356: least square misspelled
- Line 357: no "a" before prediction
- Line 430: Not a sentence.
Summary: This paper presents a promising set of methods for learning Bayesian Networks through an A* search with Lasso-based scoring function.

Submitted by Assigned_Reviewer_9

The paper considers the problem of learning sparse bayesian networks.
The paper follows the literature in addressing the problem through a dynamic programming based approach to finding the optimal ordering to determine the network
and learn the parameters of the distribution.

The contributions of the paper seem to be the use of a consistent and admissible heuristic inside A* search. The paper also proposes heuristic schemes to improve scalability of the DP A* search based approach.

The scalability gains are observable for small data sets in the numerical experiments. However, the complexity of the algorithm in the worst-case still seems exponential. So, for problems with large number of nodes in the network, the proposed algorithm doesn't seem practical. On the other hand, the SBN algorithm of Huang et al
avoids DP and guarantees to find the optimal solution in polynomial time.
The SBN algorithm is significantly slower than the algorithm in the paper.
However, the SBN algorithm comes with a polynomial time guarantee and the algorithm in the paper, though fast in experiments, is based on heuristics without guarantees.

In summary, the paper proposes fast heuristics inside a DP approach to learn sparse bayesian networks. However these heuristics don't come with any guarantees of optimality and the overall algorithm is also not guaranteed to be polynomial time.
The algorithm in the paper would be attractive if there are some theoretical results
to back up the performance of the fast heuristics in the algorithm. Also, the experiments could be more convincing when the algorithms are compared on data sets with large number of nodes.
Summary: The paper proposes a DP-based approach with fast heuristics embedded that are consistent and admissible. While the algorithm is empirically faster than state of the art, the heuristics used for scalability in the algorithm doesn't come with any theoretical guarantees.
Author Feedback

Author rebuttal: We thank the reviewers for their comments and address their concerns below.

Reviewer 1:

Regarding the comment about assuming that the distribution of the random variables is Gaussian, we believe that even with this assumption, estimating a directed Bayesian network is still a hard problem, with current optimal methods being exponential in complexity. In addition, the use of Gaussian graphical models (e.g., graphical lasso) is widely accepted for continuous variables. In the future, we can think about extending this to allow for generalized linear models. This would probably be straightforward and would simply involve replacing the LassoScore with another objective provided that the heuristic remained admissible and consistent.

With respect to the comment about further analysis of the stock market data, we wish to impress upon the reviewer that the utility of Bayesian networks is well-known, and that this class of models is widely applied to various domains that require modeling of multivariate data. In this paper, our primary goal is not to advocate for the model, but to focus on the learning algorithm. Thus, given the limited space, we have chosen to present rigorous simulations instead of a more extensive analysis of the stock market data. However, modeling the stock market is a natural application for dependency network analysis. For example, Schweitzer et al. (Science, 2009) discussed the need to understand the structure and dynamics of financial markets and Kenett et al. (PLOS ONE, 2010) emphasized the importance of networks that reveal causal relationships in determining the behavior of the stock market. Furthermore, Segal et al. (JMLR, 2010) investigated the application of a type of Bayesian networks, called module networks, to stock market data.

Reviewer 2:

To clarify how the precision/recall curves in Fig. 3 were generated, each curve was obtained from a single model estimated from data,with the best lambda according to cross validation error. Given the betas for edge strengths in the estimated model, we varied the threshold that the edge strength must exceed to be considered a positive prediction, and recorded (precision, recall) at each threshold. In order for a v-structure to be a true positive, both edges must be positive predictions and directed correctly.

Regarding the non-monotonicity in Hailfinder 2 in Fig. 3, we would like to point out that one cannot expect that precision/recall curves are always monotonic. However, it is possible that averaging over results from a larger number of simulated datasets (like more than the 30 simulated datasets that we included), will result in a more monotonic curve.

With reference to the reviewer's comment about the “random guessing line” in Fig. 3, SBN does NOT drop below the random guessing line. To clarify, the precision/recall curve for random guess is a straight horizontal line y = #positives / #cases. While this line is easy to obtain for the skeletons in Fig. 2 since it is simply a straight line at y = # true edges/# total possible edges (e.g., y = 0.1481 (52/351) for insurance, and y = 0.0773 (46/595) for mildew), it is not as straightforward to compute these values analytically for v-structures in Fig. 3. However, one can obtain these values empirically by generating random graphs over a given set of nodes and comparing them with the true graph. We performed this experiment with 50 randomly-generated graphs and found that the precision/recall curves for random guessing for v-structures are y = 0.0035 for insurance and y = 0.0020 for mildew. Thus, in Fig 3, the curves for all methods including SBN are above the random guess line.

Reviewer 3:

With reference to the theoretical guarantees of SBN, we wish to clarify that the authors of SBN do NOT guarantee an optimal solution. Although directly solving their initial formulation of the problem as a constrained optimization would result in the exact solution, the authors of SBN acknowledged that the exact solution is difficult to obtain. Instead, they sacrificed optimality by transforming the problem into a different form that can be solved with a polynomial time algorithm. We emphasize that the only guarantee that SBN provides is that the solution is a DAG under certain conditions of lambda_1 and lambda_2.

We would like to reiterate that A* lasso is the most efficient algorithm with an optimality guarantee. As we discuss in Section 2.4 of our paper, we tried inflating heuristics for which sub-optimality guarantees exist but this did not perform as well as our heuristic strategy of limiting the queue size.

The reviewer also mentioned the issue of scalability. The size of the graph that we used in simulations and the stock market data are comparable to what was used to evaluate the competing state-of-the-art methods, L1MB and SBN, in their papers. Also, the trends in Fig. 3 show that as the number of nodes in the graph increases, the improvement in performance of our method in comparison with others actually increases.